# Development and validation of School Oral Health Promotion Program "SOHEPP" oral health video for the training of adolescents and teachers in Ibadan, Nigeria

Folake Barakat Lawal[1,2,3]*, Ejiro Idiga[3], Omotayo Francis Fagbule[3], Adeola Temitope Williams[4], Okeme Ohwoka[5], Chukwuma Emmanuel Asika[6], Aderonke Adewunmi Dedeke[3], Ochuko Bright Akpobi[7], Olanrewaju Ajeigbe[8], Abimbola Muinat Oladayo[9,10], Moyosoreoluwa Dorcas Shadare[11], Omolara Moriliat Odu[12], Olayinka Egbokhare[13], Mary Ebelechukwu Osuh[1,3], Olanrewaju Ige Opeodu[1,3], Olushola Ibiyemi[1,3], Taiwo Akeem Lawal[14], Olubunmi Olusola Bankole[4], Gbemisola Aderemi Oke[1,3]

1 Department of Periodontology and Community Dentistry, University of Ibadan, Ibadan, Oyo State, Nigeria, 2 Fellow, Consortium for Advanced Research Training in Africa (CARTA), APHRC, Nairobi, Kenya, 3 Department of Periodontology and Community Dentistry, University College Hospital, Ibadan, Oyo State, Nigeria, 4 Department of Child Oral Health, University of Ibadan and University College Hospital, Ibadan, Oyo State, Nigeria, 5 Department of Oral and Maxillofacial Surgery, University College Hospital, Ibadan, Oyo State, Nigeria, 6 Department of Family Dentistry, University College Hospital, Ibadan, Oyo State, Nigeria, 7 Department of Oral and Maxillofacial Surgery, Federal Teaching Hospital, Gombe State, Nigeria, 8 Department of Restorative Sciences, Minnesota Dental Research Center for Biomaterials and Biomechanics, School of Dentistry, University of Minnesota, Minneapolis, Minnesota, United States of America, 9 Department of Oral Pathology, Radiology and Medicine, College of Dentistry, University of Iowa, Iowa, Iowa, United States of America, 10 Iowa Institute of Oral Health Research, University of Iowa, Iowa, Iowa, United States of America, 11 Department of Cosmetic Techniques and Management, Faculty of Business, Durham College, Ontario, Canada, 12 Department of Adult Education, University of Ibadan, Ibadan, Oyo State, Nigeria, 13 Department of Communications and Language Arts, University of Ibadan, Ibadan and Biomedical Communications Centre, College of Medicine, University of Ibadan, Oyo State, Nigeria, 14 Division of Pediatric Surgery, Department of Surgery, University of Ibadan and University College Hospital, Ibadan, Oyo State, Nigeria

* folakemilawal@yahoo.com

## Abstract

Presently, in Nigeria, as in many other developing countries, oral health promotion programs remain the responsibility of the public health dentists. This responsibility is now being threatened, particularly with persistent short supply of dentists in general. Recent evidence showed that peers and school-teachers trained by dentists can effectively anchor oral health promotion programs. However, the sustainability of this method of oral health promotion is further threatened by the emigration of dentists needed to train these peers and teachers. Therefore, the assessment of the feasibility of a training alternative, such as the use of pre-recorded video, has become crucial. This study reports the development and validation of an animated video as an oral health training tool for adolescents and teachers in Ibadan, Nigeria. The narrative video was developed by a team of experts from various specialties, which included

**Data availability statement:** Transcripts are available as a supplementary file.

**Funding:** This research was supported by the Consortium for Advanced Research Training in Africa (CARTA). CARTA is jointly led by the African Population and Health Research Center and the University of the Witwatersrand and funded by the Carnegie Corporation of New York (Grant No. G-19-57145), Sida (Grant No:54100113), Uppsala Monitoring Center, Norwegian Agency for Development Cooperation (Norad), and by the Wellcome Trust [reference no. 107768/Z/15/Z] and the UK Foreign, Commonwealth & Development Office, with support from the Developing Excellence in Leadership, Training and Science in Africa (DELTAS Africa) programme. The statements made and views expressed are solely the responsibility of the Fellow."

**Competing interests:** The authors have declared that no competing interests exist.

public health dentists, pediatric dentists, epidemiologists, communication scientists, animation experts, dental surgery residents, a pediatric surgeon, and science educators. The steps involved in the development included pre-video production research, development of the key oral health education messages, development of the script and storyboarding, creation, design, and selection of characters, visual style decision, voice-over narration recording, production, and post-production processing. The script for the video was developed from information adopted from a manual on oral health care of students. The video produced was validated for face and content quality among dentists, teachers, and adolescents in secondary schools through individual feedback and focus group discussions. Relevant comments and observations were considered in the final editing and postproduction process to yield a final video after series of editing. The video was found to be simple to use, clearly illustrative and highly educative. This is a report on the development, production, and validation of the School Oral Health Promotion Program animated video for training adolescents and teachers.

## Introduction

Oral health conditions and practices vary as an individual journey through the different stages of life, such as adolescence, typically defined by the World Health Organization as the period of life between ages 10 and 19 years [1]. Adolescence is characterized by pronounced social and psychological vulnerabilities, which can profoundly impact an individual's well-being and development [2]. In terms of oral health, this period represents a time of transition into adulthood and the last opportunity to influence oral health behavior and status. In addition, adolescence has been associated with high dental disease risks such as caries, increased risk for traumatic injury and periodontal disease, a tendency for poor nutritional habits, potential for drug abuse, pregnancy, eating disorders and an increased desire for esthetics [2,3]. Amongst all of these, dental caries, gingivitis and periodontal disease have been the most commonly reported oral diseases in this age group [2,4]. The prevalence of dental caries peaks during adolescence and has been associated with environmental factors including dietary habits, autonomy in seeking or avoiding care and a low priority for oral hygiene practices [3]. Untreated dental caries in this age group can manifest as pain and absenteeism [5] from school due to related discomfort and may exacerbate vulnerabilities to substance abuse, which adolescents are already predisposed to. Similarly, adolescents in Nigeria have been shown to have a high prevalence of gingivitis [4–6]. In addition, they have also been shown to have a high level of unmet dental needs and with low level of dental utilization, compounding this issue, these adolescents also exhibit a low level of oral health awareness [5,7].

Meanwhile, the adoption of appropriate personal oral hygiene and regular professional intervention can prevent the occurrence of these conditions and their adverse effects. It is, therefore, imperative to implement preventive strategies aimed at promoting oral health in this special category of individuals and one of such strategies is

the execution of Oral Health Education (OHE) programs [5]. These programs have been shown to be effective in influencing good oral health behavior, as they equip individuals with appropriate knowledge on the prevention of diseases, bring about a change of attitude and provide them with adequate skills to change their negative health practices [8–11]. Several tools for health education exist such as didactic lectures, posters, songs and audiovisuals [8–11]. Among these, the use of audiovisuals has the capacity for a longer-lasting effect because it dramatizes the message and brings a sense of realism, which can have a great impact on its audience [9]. However, there is a dearth of literature on the development of an audiovisual tool that is culturally adapted and contextually appropriate for our environment in Nigeria to improve the oral health of adolescents, who constitute about one-quarter of the population.

Furthermore, teachers have also been shown to be effective channels of educating children and adolescents especially in this era of shortage of trained dental personnel to carry out the OHE programs in schools [12]. Given that adolescents spend a substantial portion of their day in school under the supervision of teachers, equipping teachers and adolescents in school with adequate knowledge and resources can enable them to effectively promote oral health. By integrating oral health education into the curriculum or through dedicated educational sessions, teachers can effectively impart crucial information on proper oral hygiene practices, dietary habits conducive to good oral health and other positive oral health behaviors in adolescents, thereby contributing significantly to their overall well-being. Additionally, teachers can serve as role models by demonstrating good oral hygiene practices themselves and by creating a supportive environment that encourages students to prioritize their oral health. Although previous studies [13,14] have reported willingness of teachers to contribute to oral health promotion in schools, there is a paucity of data on the development of any tool to equip them with the adequate knowledge and skills necessary to promote oral health amongst adolescents.

Therefore, to promote good oral health and reduce the prevalence of oral diseases in this age group, this study was designed to develop and validate a narrative oral health education video for the training of adolescents and teachers in Oyo State, Nigeria.

## Materials and methods

This was a methodological study that described the process of development and validation of an educational video focused on improving oral health in the school settings.

### Ethics statement

Ethical approval was granted by the Joint University of Ibadan/University College Hospital Ethics Review Committee. Written informed consent was obtained from all students, parents and teachers involved in the study. Assent was also obtained from the students. The study (development and validation of the video) was conducted between 20 October 2022 and 31 January 2023.

### Development and production of the video

#### Background

The video named "School Oral Health Promotion Program" (SOHEPP), a 25-minute animated video, was developed in English, and then translated into Yoruba and Pidgin English languages. The video was developed with the aim of providing an alternative form of oral health training for teachers and adolescents in secondary schools to bridge the gap created by the sparse number of dentists available in Nigeria and other developing countries to perform the role of physical trainers. The development of the video was based on the Research and Development model (R&D) [15] and Community Based Participatory Research (CBPR) [16–18]. The R&D model for educational product is a systematic process of investigating the health needs of a target population through research to create new educational products, processes, or technologies and bring them from research stage into production [15]. The ten principles of R&D (Table 1) include Research and Information Collection, Planning, Developing Preliminary Form of Educational Product, Preliminary Field Testing,

**Table 1. The R&D principles and process of the development, production, and validation of the video.**

| Principle | Activity/process involved | Stage in development/validation of the animated video for the study |
|---|---|---|
| 1. Research and Information Collection | Need analysis | Pre-video production research |
| 2. Planning | Formulating skills and expertise regarding the problem of the research. | Development of the key oral health education messages in the video |
| 3. Developing Preliminary Form of Product | The preliminary educational product, is developed by preparing and evaluating the supporting components | Creation, design, and selection of characters. Visual style decision. |
| 4. Preliminary Field Testing | The preliminary product is evaluated among some selected parties (3–4) through interview, questionnaire or observation to and analyze the data for next step | Recording of voice-over narration Production of the video **Validation** Face and content validation by experts. Face and content validation by target population - teachers and the adolescents. |
| 5. Revising Main Product | The preliminary product is revised using the data obtained in step four. | Revision and editing of the video. Face and content validation by experts. Face and content validation by target population - teachers and the adolescents. Validation among adolescents on a large scale. |
| 6. Main Field Testing | The revised educational product is evaluated among selected participants (5–15) by qualitative method. | |
| 7. Revising Operational Product | The revised product in this step is revised based on the data obtained and developed as an operational model design to be validated among a larger population. | |
| 8. Operational Field Testing | The validation of operational educational product is conducted among larger number of population (30–40) through interview, observation, or questionnaire to obtain data for revising the final product. | |
| 9. Revising Final Product | The product is completely revised by the gained data in step eight and launched as the final educational product. | |
| 10. Dissemination and Implementation | The product dissemination is conducted through seminars, publication, or presentation to related stakeholders. | Ongoing evaluation of the video on oral health among adolescents in schools |

Revising Main Product, Main Field Testing, Revising Operational Product, Operational Field Testing, Revising Final Product, and Disseminating and Implementing the educational product.

**Stages in the development and production of the video.** The stages involved in the development and production of the video (Table 2) included Stage one; Pre-video production research, Stage two; Development of the key oral health education messages in the video, and Stage three; development of the script and storyboarding. Other stages included creation, design, and selection of characters, visual style decision, recording of voice-over narration, production of the video and validation of the video [16–19].

**Stage one: Pre-video production research.** The pre-video planning included focus group discussions (FGDs) conducted among 201 adolescents [20]. The FGD was aimed at exploring perspectives about the use of video as a

**Table 2. Stages in the development and production of the video.**

| Stage | Development process |
|---|---|
| One | Pre-video production research |
| Two | Development of the key oral health education messages in the video |
| Three | Development of the script and storyboarding |
| Four | Creation, design, and selection of characters |
| Five | Visual style decision |
| Six | Recording of voice-over narration |
| Seven | Production of the video |

training tool for promoting oral health among adolescents. The FGD was also deployed to assess the adolescents' self-perceived oral health knowledge needs that would be included as content of the video. The involvement of major stakeholders in the development and validation of projects is a good foundation for a successful intervention [18]. The FGDs involved 201 adolescents because the initial discussions with the students from some schools elicited sparse responses due to fear of dentists from the students as dentists had not previously visited the schools prior to the FGD. Thus, the researchers had to continue with the FGDs until saturation was reached.

The FGDs showed that "the adolescents were enthusiastic about the use of video as a training tool and would want it to include details of causes of oral diseases and how to care for the teeth" [20].

**Stage two: Development of the key oral health education messages in the video.** The video was developed by a team of experts from various specialties including public health, pediatric and general dentistry, epidemiology, communication, animation, oral and maxillofacial surgery, and science education. The oral health information content of the video were adapted from extensive review of literature and from manuals for the oral health of school students and training of teachers [21–23], which was modified for the Nigerian environment. The video portrays basic knowledge of oral health and hygiene, the importance of prevention of oral diseases, as well as the roles of diet, and smoking in oral health. It also illustrates how to recognize symptoms and indicators of poor oral hygiene status such as obvious plaque, calculus, and tooth decay. In addition, it teaches skills on effective tooth brushing habits.

**Stage three: Development of the script and storyboarding.** The contents and key messages on oral health and hygiene were written out from the manual and shared with the communication experts and professional scriptwriter who transformed this into a script. Thereafter, the script was shared among all the members of the team from the various specialties as earlier identified, for their objective assessment, comments, fact-checking and other inputs [17,19].

The plot of the video revolves around two adolescents who had planned to go out to the playground to play football. The first scene showed *Seun* holding a ball in his hands and walking into *Ade's* residence (S1 Appendix). His mission was to invite his friend to come out to play football. *Ade* was shown sitting on a flight of steps in the backyard with a swollen jaw. He has his palm pressed on his right cheek and his head bent. He was wincing in pain. He narrated his experiences with toothache over the preceding days. He described the sleepless nights he endured, the ineffectiveness of medications and the various harmful substances he had tried such as alcohol, bitter leaf (Vernonia amygdalina, a vegetable widely eaten in some parts of Nigeria), salt, and "car battery water" (distilled or deionized water used to refill lead-acid batteries), which only worsened his condition. He then shared his anxieties and fears about visiting the dentists. This led to a discussion between the two boys during which *Seun* recounted his mother's recent experience with toothache, which was relieved after she visited a dentist. He also shared his personal and his sister's pleasant experiences with the dentist (S2 Appendix). The scene ended with *Seun* searching for and finding an oral health video for them to watch and get better educated.

The second scene featured the oral health video that *Seun* brought out for his friend to watch. This began with an interview of *Dr Fresh*, a female dentist (S3 Appendix). The scene opened with the dentist discussing the basic anatomy of the teeth, functions of the teeth, common oral problems, and causes, prevention, and treatment of common oral conditions. In addition, there was a scene that described the tooth-cleaning process using a toothbrush on a jaw model.

In the third scene, the focus returned to the two boys with *Ade* chastising *Seun* for not sharing the video with him earlier, as the information could have prevented his toothache. *Seun* then summarized some aspects of the video while reiterating the need to share the video with others. The scene ended with a song (music) by one of the authors on good oral health practices using the beats of the soundtrack of a song that was popular among adolescents in the environment, which was downloaded from a public domain.

Language experts checked the script for correctness and translated it into Pidgin English (a widely spoken English-based creole in Nigeria and Yoruba (a major indigenous language spoken in southwest Nigeria). An independent linguist

back translated the scripts for the two versions: Pidgin English and Yoruba. The back-translated script had very little differences from the original version. Afterwards, the animator began working on character creation.

**Stage four: Creation, design, and selection of characters.** After the animator created and designed the characters for the video; both human and animated characters, based on the script, three adolescents were asked to, independently, select the most appropriate character. All the adolescents chose the animated character, expressing that they preferred this form of representation. Thereafter, the animator created four characters that differed by age for the three adolescents to pick from as the most preferred character. The adolescents picked late-adolescent characters (S1 Appendix) as the most appropriate for them. This was later shown to the research team to appraise [17,19], based on the consideration of the profile of the target population, the proposed characters were accepted after thorough deliberations.

**Stage five: Visual style decision.** The animator created the visual style frame and sent it to the project team for assessment and comments. This included the art design of the entire video, including illustrations, characters, images, color palette, typography, and keyframes. The project team deliberated on the art design, and the animator modified it accordingly based on the team's suggestions [17,19].

**Stage six: Recording of voice-over narration.** After the selection of the animation characters, the media experts commenced work on the voice over recording of the script in conjunction with other members of the team. The voices were tested, and some were selected by the communication experts and the animation artists. Thereafter, the scripts were given to the selected voice artists to read and get familiar with, in preparation for rehearsals. The rehearsals for the different languages commenced soon after this with the media experts coordinating. In addition to the group rehearsals, actors were encouraged to rehearse on their own in the comfort of their homes to engender proper assimilation of the scripts. The final rehearsal was conducted a day before the final recording of voices with some members of the project team to ensure correct pronunciation and adherence to the script. The final recording of voices was done in the recording studio, to ensure fidelity of sound and a great audio output.

**Stage seven: Production of the video.** After the recording of the voice over narration and the approval of the characters and visual style by the project team, the animator continued with the production process. This included further work on illustrations, animation, and other aspects of production, achieved through rigging, keyframes creation, interpolation and reviews by the media and creative team. Rigging is a process in animation where the skeletal structure that controls the movement of each part of illustration at specific points in time is done [24]. Interpolation is a step during animation creation where smooth movement of the characters is achieved with the aid of some designated software [24]. These were followed by an audio mix to edit and synchronize the audio recording with the visuals (synching of characters, pictures, and flow with sound). The media team finally integrated all the components and rendering was carried out by combining the digital assets into a single video file. The delivery of the video was completed by the media team and the video was shared with the entire project team (media and research team members) who scrutinized the video and made useful suggestions and edits, which were incorporated into the final version.

## Validation of the video

The video was validated in three stages (Table 3) for content and face (appearance) validity.

**Stage one: Face and content validity of the video by non-project experts.** After the video was accepted by the project team, the Pidgin English version was first shared independently among four non-project members who were dentists to assess the validity of the content of the video, its quality in terms of clarity, brightness, or dullness (contrast), pictures/illustration quality, appropriateness of the scenes and other general evaluations. The comments from the dentists were incorporated and the video underwent further editing before it was shared with the teachers and students.

**Stage two: Face and content validity of the video by stakeholders (teachers and students).** This step is a form of preliminary field testing of the video among stakeholders and revision of the video under the R&D principle. For this stage, the video was sent to eight teachers including school administrators such as the vice-principal in four different schools for

**Table 3. Process of the validation of the video.**

| Stage | Validation of the video |
| --- | --- |
| One | Face and content validity of the video by non-project experts |
| Two | Face and content validity of the video by stakeholders (teachers and students) |
| Three | Validation of the video among a larger number of teachers and students through FGDs |
|  |  |

assessment. The importance of feedback from stakeholders in video validation has been reported [16–19]. The project covered the cost of internet data needed by the teachers to download the video on their phones, share the video with others if they deemed it fit and for feedback on the evaluation of the video. The responses of the teachers to questions were obtained via WhatsApp messaging and phone calls to minimize disruption of school activities and to increase the probability that the video would be watched at their convenience. The questions asked after the teachers had watched the video included the appropriateness of the video as a training tool for teachers and students, what was good about the video, what should be modified, corrected, removed, or added to it.

The comments of the teachers were noted. Thereafter, the video was also shown to a set of four students in four different schools with their comments on the same questions that the teachers were asked with an additional question for them to summarize what they have learnt from the video. The responses of the adolescents were obtained independently. The comments and feedback received from the teachers and adolescents were deliberated upon by the project team and sent back to the animator for further editing on all the versions of the video.

**Stage three: Validation of the video among a larger number of teachers and students through FGDs.** The FGDs were conducted after the English and Yoruba versions of the video were shown to teachers and students in three other schools different from the previously selected schools (where the pidgin version was shown). One FGD was conducted for teachers (as data saturation had been reached), and additional FGDs were held for students. The purpose was to further confirm and triangulate the responses of both teachers and students.

## Data analysis

The responses of the teachers were summarized, and themes were generated (S1 Text). The audio-recording of the FGDs were transcribed verbatim. The script was also read through while listening to the audiotape recording to ensure that the transcribed data conformed with the audio recording and errors noted were corrected. Two independent trained researchers compared the accuracy of the transcripts with recordings of the discussions on the audio tapes. After this, it was read over multiple times to identify themes. Codes were generated and applied to the transcripts. Thereafter, another member of the research team applied the codes to the transcripts independently. New codes generated were added to the already existing ones. The analyst' triangulation was done in such a way that the two independent researchers who were epidemiologists applied the codes to the transcripts and these were compared. Further analysis of data was based on thematic induction.

**Trustworthiness of Data.** Based on the criteria of trustworthiness to ensure rigor and quality of qualitative data, four criteria have been identified: credibility, transferability, dependability and confirmability [25–28].

**Credibility.** The interview and focus group guides were developed by the public health dentists and epidemiologists with experience in qualitative research. The questions were developed through extensive literature review of video validation process. In addition, the moderator of the interviews and discussions was trained in facilitating interviews and focus group discussions. The students were selected using purposive sampling technique and students from different

socioeconomic classes were selected to ensure diversity of participants and robust data. The summary of the responses was done to ensure the responses of the participants were well captured. This was to minimize bias and ensure correctness of data captured at the interviews and discussions.

**Transferability.** The details of the interviews and focus group discussions, data collection and analysis have been provided for ease of placing the results in context thereby aiding comparability of the finding with other studies.

**Dependability.** The process of conducting the research has been well described to enable replication by others.

**Confirmability.** The adolescents that participated in this study were from the age range 10–19 years, thus from a homogenous population with the same experiences. In addition, data for the validity of the video was collected from multiple schools within the city to enhance triangulation of the data, enriching the robustness of the data.

## Results

### Sociodemographic characteristics of the teachers and adolescents that commented on the animated video

A total of eight teachers participated in the WhatsApp and phone calls feedback with mean age of 44.1 ± 8.2years. There were four females and two of the teachers were vice principals and their years of experience ranged from 5-26years. Only four of the teachers had comments on the video, others were satisfied with the video.

For the FGDs, 12 teachers participated with a mean age of 46.8 ± 4.9years while a total of 60 adolescents with age range of 10–19 years participated in the focus group discussions with equal gender distribution and the mean age was 13.6 ± 2.3years.

The summary of the findings using thematic analysis from the WhatsApp messages from the teachers is described below.

**Language.** A major comment by the teachers who watched the pidgin version of the video was on the need for a translation of the video into other languages for wider coverage and ease of communication with the students. Below are some quotes from the teachers.

*"Also, if a step can be taken further to translate the message into three common Nigerian languages. This will enhance widest dissemination of the message." (Male)*

*"I will love it if it can be performed in other languages, e.g., Advanced English, Yoruba, Hausa, etc (Female)*

*"The information can be developed for less-literate individuals for them to know the tooth problems and how they can go about it instead of using battery water" (Female, Vice principal)*

The teachers appreciated the video, below are the themes and some of the quotes from the teachers to support the themes.

### Presentation of the video

*"The video was well composed and arranged, presentation cuts across ages and the message was clearly explained".*

*"It is highly educative and informative"...*

*"The video is presentable for everyone as most of the information needed are there." (Female, Vice Principal)*

*"The causes of the tooth problems were well explained there". (Female, Vice Principal)*

*"It is well explanatory and super educative". (Female)*

*"The message of the video is well passed across. It is enlightening". (Female)*

*"The content of the video is incredible". (Female)*

*"It educates and informs people about caring for their teeth and mouth (Female, class teacher)*

**Appropriateness**

*"It is appropriate for both the teachers and students".*

*"…………. presentation cuts across ages"*

**Dissemination**

The teachers mentioned ways of disseminating the video to the target audience and wider community.

*"To have widest coverage of audience, do you permit it being forwarded to various platforms and groups one belongs to?".*

*"Schools with projectors can even project it on a screen to make the message permanent in the students' memories".*

The summary of the findings from the adolescents after watching the pidgin version of the video is as described below. **Tooth cleaning aids.** A major comment by the adolescents on what they did not like about the video was that the specific toothpaste to be used was not mentioned in the video as quoted below.

*"I think that in the video they are supposed to put a special Maclean or what can be used….….and the kind of toothbrush."*

**Music.** A major comment that the students who only watched the pidgin version of the video was that the music which came up at the end of the video was short and they wanted it to be longer.

*"I also like the music, add more and let the music be longer."*

**Language**

Some of the students preferred the Pidgin version, while some wanted translation of the video into Yoruba and English

*"… It should be translated to English."*

*"……like some of us do not understand the language so we prefer it the local language."*

**Pictures of tooth decay**

Some of the students did not like pictures of the oral diseases displayed in the video as opined by some that it was irritating.

*"The area I don't like is where they showed us the teeth that decay, rotten teeth"* It is irritating.

However, some believed it is a source of reminder to carry out what was recommended,

*"I don't think that you should remove the pictures, because if they show it to others, they will understand that it is our attitude, if they are doing that it is not good. So, I don't think that they should remove it."*

**Editing and revision of the video after the initial comments of watching the pidgin version of the video**

The comments about what was not liked about the video were noted and utilized to improve it. Suggestions included extending the music at the end of the video and specifying that the toothpaste must contain fluoride, although no brand name was mentioned. Also, the type of toothbrush to be used was also emphasized, although no brand of toothbrush was mentioned. Also, improvement on the sharpness and color contrast of the video was suggested and improved upon. All other comments of the teachers and adolescents were noted, and the video was edited until all the relevant comments of the teachers and adolescents were treated. and those who shared the comments were satisfied.

**Themes emerging from the feedback from the adolescents after watching the three versions of the video**

**Appropriateness of the video for students.** All the students mentioned that the video was appropriate for them, here are some excerpts:

*"Yes, it is okay for us. It is perfect for our age group because some of us do not know how to brush our teeth, so, it teaches us more on how to brush and be careful with our gum".*

*"It is appropriate for our age because of the language they used."*

*"It is appropriate for us because it is in the form of um… animation, it is a form that is attractive for us to watch, we want to know everything that is going to happen."*

**What the students liked about the video**

**Content of the video.** The students liked the content of the video. For example, a student said ….

*"When they are talking about some diseases, which young people don't know about, like periodontal diseases."*

*"…. because of the way they showed us how to brush our teeth, because of the information that was given to us".*

**Character.** The teenage characters in the video were also appreciated.

*"I like that they use teenagers, because the teenagers express themselves well and the pictures they used when they were explaining the way of treating the teeth and disease so that was like."*

**Language used in the video.** Some were impressed with the pidgin language.

*"…...the language that they used, like it is even making me more entertained".*

**The interaction between Dr Fresh and the presenter.**

*"When Dr. Fresh and the presenter were talking, the way he was interacting with the doctor. The way the doctor explained everything about the mouth odor and the diseases".*

**Animated human character.** The adolescents reported that the animated human characters were appropriate,

"*For young children, this cartoon is okay but for adults the human form is the best for them.*

"*People prefer cartoons to drama they will enjoy it the more if they leave it the way it is.*"

**Music.** The adolescents also liked the musical content, which was used to conclude the video.

"*I liked the video because of the music and the character.*

**Length of the video.** The adolescents mentioned that the length of the video was adequate.

"*It is not too long and not too short because it has all the necessary information, so there is no need to add any more content and there is no need to cut things out of it, so it is okay like this*".

"*I think it is okay because in as much as it contains all the appropriate information we need to know about the teeth.*"

**Summary of things learnt.** All the adolescents mentioned several things they learnt from the video. Below are some quotes from them.

"…………*we should brush, you brush: there are steps to take when brushing, we should avoid sugary foods and unbalanced diet.*"

"………………*they showed a dialogue program which um…showed us the proper hygienic way of taking care of our teeth, brush it every day*".

"………*in the video they showed us the amount of toothpaste we are to use, some people spread it on the brush, and it is supposed to be pea size*".

"*I also learnt that if we want to brush our teeth, we should be extra careful when brushing it to avoid blood coming out from the gum*".

"*I learnt that after we drink anything that has acid, it should be 30 minutes before we brush our teeth*".

**Further editing and revision of the video.** Editing and revision of the video was carried out until (there were no more comments from the teachers and adolescents who shared their reservations after watching the video) they were all satisfied.

## Discussion

This study described the process of the development, production and validation of an animated video based on a modification of the R&D model and the CBPR. The video was liked by the adolescents and the teachers. It was also found to be simple to use, clearly illustrative and highly educative. The teachers recommended its wide circulation via the social media platform.

The study was not without limitations; the validation process was based on qualitative methods without quantitative methods input, which could have enabled the quantification of the responses, however, the importance of the participants views is an important concept in R&D model. Furthermore, participants' views and perspectives were most likely captured using qualitative methods.

The process of development of the video began with sourcing for evidences from previous researches [20], which guided the content of the video. The development of the key messages and contents of the video, script, and

storyboarding, were notable stages, which included experts in oral health, language, and communication arts. This was important in ensuring the quality of the content and the information to be passed across to the target audience. Script development and storyboarding have been reported by authors as a critical stage in the development of educational tools to promote oral and general health [29]. Further, the creation, design, and selection of characters preceding the final production of the video reflected great involvement of the major stakeholders, in this case the adolescents in addition to the project team. This can be described as co-production interaction and the contribution of this process to the effectiveness of videos have also been reported [19].

The process of the production of the animated video also involved experts experienced in health education videos and tools to ensure optimum quality of the video output [30,31]. Use of appropriate specialists in video development and production are contributory factors to smooth production, delivery and post production processing of the video [25].

The validation process of the video included its appearance (face) and content validity utilizing feedback from major stakeholders [15,17,18]. One of the feedback comments from the teachers had to do with the language used for producing the video content with a suggestion that the three major Nigerian languages be adopted in reproducing the video for a wider reach. The role of language as an important contextual factor in the understanding of the content of educational materials has been reported [32]. Additionally, reports from around the world highlight the importance of language in the creation of educational materials [33].

The students identified music as an important aspect of videos, this is not surprising as music has been used as an educational tool for promoting health and has been similarly documented in previous studies [34,35].

The immediate evaluation of the effectiveness of the video on knowledge was observable in the succinct summary of things learnt by the adolescents. This finding is consistent with previous studies, which showed that videos are a means of edutainment among adolescents, thus effective in learning new things [36,37]. The immediate evaluation of the video on knowledge is encouraging and led to an extra step of evaluating the effectiveness of the video among adolescents at an intermediate evaluation period. In addition, this finding is a pointer that the video can be rolled out by the ministry of education for large coverage.

## Conclusion

This is a report on the development, production, and validation of an oral health video aimed at training adolescents and teachers about their oral health. The report described in detail the processes involved in the production of the final SOHEPP animated video, part of a larger study. The video, liked by the adolescents and the teachers, was found to be simple to use, clearly illustrative and highly educative. Based on the satisfactory comments on the three versions of the video, they have been made available for the adolescents and teachers to watch and utilize in training themselves and others. Given the positive reception of the oral health video by adolescents and teachers, policymakers could consider incorporating such multimedia tools into school curricula as part of a standardized oral health education program.

## Supporting information

**S1 Text. Transcript of the qualitative data.**
(DOCX)

**S1 Appendix. *Seun* holding a ball in his hands and walking into the compound of the house of *Ade.***
(DOCX)

**S2 Appendix. *Seun* recounting experiences with the dentist.**
(DOCX)

**S3 Appendix. Interview of *Dr Fresh*, a female dentist.**
(DOCX)

## Acknowledgments

The teachers and students who participated in the study are acknowledged for their roles.

## Author contributions

**Conceptualization:** Folake Barakat Lawal.

**Data curation:** Folake Barakat Lawal.

**Formal analysis:** Folake Barakat Lawal, Ejiro Idiga, Omotayo Francis Fagbule, Adeola Temitope Williams, Aderonke Adewunmi Dedeke, Abimbola Muinat Oladayo, Olayinka Egbokhare, Mary Ebelechukwu Osuh, Olanrewaju Ige Opeodu, Olushola Ibiyemi, Taiwo Akeem Lawal, Olubunmi Olusola Bankole, Gbemisola Aderemi Oke.

**Funding acquisition:** Folake Barakat Lawal.

**Investigation:** Folake Barakat Lawal, Ejiro Idiga, Omotayo Francis Fagbule, Adeola Temitope Williams, Okeme Ohwoka, Chukwuma Emmanuel Asika, Aderonke Adewunmi Dedeke, Ochuko Bright Akpobi, Olanrewaju Ajeigbe, Abimbola Muinat Oladayo, Moyosoreoluwa Dorcas Shadare, Omolara Moriliat Odu, Olayinka Egbokhare, Mary Ebelechukwu Osuh, Olanrewaju Ige Opeodu, Olushola Ibiyemi, Taiwo Akeem Lawal, Olubunmi Olusola Bankole, Gbemisola Aderemi Oke.

**Methodology:** Folake Barakat Lawal, Ejiro Idiga, Omotayo Francis Fagbule, Adeola Temitope Williams, Okeme Ohwoka, Chukwuma Emmanuel Asika, Aderonke Adewunmi Dedeke, Ochuko Bright Akpobi, Olanrewaju Ajeigbe, Abimbola Muinat Oladayo, Moyosoreoluwa Dorcas Shadare, Omolara Moriliat Odu, Olayinka Egbokhare, Mary Ebelechukwu Osuh, Olanrewaju Ige Opeodu, Olushola Ibiyemi, Taiwo Akeem Lawal, Olubunmi Olusola Bankole, Gbemisola Aderemi Oke.

**Project administration:** Folake Barakat Lawal, Taiwo Akeem Lawal.

**Resources:** Folake Barakat Lawal, Taiwo Akeem Lawal.

**Supervision:** Folake Barakat Lawal, Gbemisola Aderemi Oke.

**Validation:** Folake Barakat Lawal, Ejiro Idiga, Omotayo Francis Fagbule, Adeola Temitope Williams, Okeme Ohwoka, Chukwuma Emmanuel Asika, Aderonke Adewunmi Dedeke, Ochuko Bright Akpobi, Olanrewaju Ajeigbe, Abimbola Muinat Oladayo, Moyosoreoluwa Dorcas Shadare, Omolara Moriliat Odu, Olayinka Egbokhare, Mary Ebelechukwu Osuh, Olanrewaju Ige Opeodu, Olushola Ibiyemi, Taiwo Akeem Lawal, Olubunmi Olusola Bankole, Gbemisola Aderemi Oke.

**Writing – original draft:** Folake Barakat Lawal.

**Writing – review & editing:** Folake Barakat Lawal, Ejiro Idiga, Omotayo Francis Fagbule, Adeola Temitope Williams, Okeme Ohwoka, Chukwuma Emmanuel Asika, Aderonke Adewunmi Dedeke, Ochuko Bright Akpobi, Olanrewaju Ajeigbe, Abimbola Muinat Oladayo, Moyosoreoluwa Dorcas Shadare, Omolara Moriliat Odu, Olayinka Egbokhare, Mary Ebelechukwu Osuh, Olanrewaju Ige Opeodu, Olushola Ibiyemi, Taiwo Akeem Lawal, Olubunmi Olusola Bankole, Gbemisola Aderemi Oke.

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
