## [Decision Letter · Decision Letter 0]

4 Dec 2024

PGPH-D-24-02493

Development and validation of School Oral Health Promotion Program “SOHEPP” oral health video for the training of adolescents and teachers in Ibadan, Nigeria

Dear Dr. Lawal,

Thank you for submitting your manuscript to PLOS Global Public Health. After careful consideration, we feel that it has merit but does not fully meet PLOS Global Public Health’s publication criteria as it currently stands. Therefore, we invite you to submit a revised version of the manuscript that addresses the points raised during the review process.

Please address my 15 comments listed below in addition to those of the two reviewers.

We look forward to receiving your revised manuscript.

Kind regards,

Maha El Tantawi

Academic Editor

Additional Editor Comments (if provided):

The paper provides useful information on the development of a video for oral health education. The manuscript (MS) can benefit from some organization to help readers better follow its flow. Please address the following:

1- Please revise the MS for English language including mainly grammar such as, for example but not limited to: L64 (an individual journeys)- singular plural mismatch, L72 (oral disease)- this should be in the plural. Please also revise for syntax and flow, like L107 (In addition, the importance of oral to general health has been reported by adolescents in this environment)- this sentence oes fit here and interrupts the flow which was talking about health education. Please, remove it. Please mention the full term for SOHEP the first time it i introduced in Abstract then also again in the MS. Also, divide the text into paragraphs to help readers follow. For example, it may be advisable to begin new paragraphs in Intro L81, L88, and so on.

2- Later (L157), it becomes clear that the 25 minute video is like a story with several scenes (maybe 3). It would be helpful if this is clarified in the Abstract (L47) and earlier in the MS (L117) by using a term to describe the video like story telling or narrative or something similar to allow readers to follow. Until I reached L157, I had no idea whether the video was a lecture, interviews with people with oral health problems or a story.

3- Also, the term "animated" is nor very clear. This can be addressed by adding an appendix that includes screenshots of the video to help readers understand. I am not sure if - based on the description of the video in the MS- I would see actors moving and talking in the video or cartoons or other kind of moving characters. Screenshots may also help clarify what is meant by (L197, "illustrations, characters, images, color palette, typography, and keyframes") and what the differences among them are.

4- In addition to the screenshots, it is important to add two separate figures. The first shows the steps of the video development and the second shows the steps of validation. The terms describing the steps in the figures/ illustration should match the subheading in the text elaborating on the steps. In the text, please use numbering to show which steps fit under (1- Development) and (2- Validation). For example, L351 mentions "three versions of the video". What are they? Where do they fit in the development and validation steps?

5- Abstract: In Methods, please briefly describe the steps of development and then of validation addressing the how, and by who of both. Please consider adding to the keywords the term "video".

6- Introduction: Please, support the sentence ending in L93 by a reference.

7- Methods: L122, please briefly clarify what the R&D model means. Please also in the same line clarify what the principles are. They may be listed afterwards so maybe numbering them in sequence in the paragraph may help readers follow.

8- Methods: Why are 201 adolescents needed for the FGDs? This is a very large number for FG. Saturation must have occurred much earlier than by the time all were included. How many groups were they split into? How long did the meeting with each group take?

9- Methods L184: Why were revisions needed if back translation showed no differences.

10- Methods L213: What does "rigging" and "interpolation" mean?

11- Methods- L232 "responses": Please clarify what the responses were for. Where there close ended questions? a standardized feedback form? or were they asked something like "how did you like the videos"?

12- There is no Results section. Starting L242 "A major comment...." looks like and are Results. Please, put them under the heading of Results.

13- L322-323 "For young children, this cartoon is okay but for adults the human form is the best for them.": How is this music as indicated by the subheading?

14- The authors describe the study and data as qualitative. Which qualitative analysis techniques were used for analysis? This needs to be clarified in Methods. Also, in the data availability statement, the authors indicated there is no dataset for this qualitative study. There should be. I assume these are the transcripts that the authors are quoting in the last section of the MS although it is important to clarify how they were obtained and analyzed as indicated in my comment.

15- There is no Discussion section. Qualitative studies and studies developing and validating products do and need Discussion. Please use the STROBE guidelines to develop the Discussion.

Reviewers' comments:

Reviewer's Responses to Questions

**Comments to the Author**

1. Does this manuscript meet PLOS Global Public Health’s publication criteria ? Is the manuscript technically sound, and do the data support the conclusions? The manuscript must describe methodologically and ethically rigorous research with conclusions that are appropriately drawn based on the data presented.

Reviewer #1: Yes

Reviewer #2: Yes

2. Has the statistical analysis been performed appropriately and rigorously?

Reviewer #1: No

Reviewer #2: N/A

3. Have the authors made all data underlying the findings in their manuscript fully available (please refer to the Data Availability Statement at the start of the manuscript PDF file)?

Reviewer #1: No

Reviewer #2: No

4. Is the manuscript presented in an intelligible fashion and written in standard English?

Reviewer #1: Yes

Reviewer #2: Yes

5. Review Comments to the Author

Reviewer #1: The manuscript is suitable for publication. The study lacks quantitative data and is well-written, requiring minimal modifications. I have also attached the reviewer's comments so the authors are aware of the corrections they need to make.

Reviewer #2: Overall, the paper describes the process undertaken for the development of a training video on dental health aimed at adolescents. It is a qualitative analysis only and limited to 4 dentists, 4 teachers and 4 adolescents each. Attempt could have been made to quantify some of the comments against a prestructured questionnaire. Also, the authors do not describe whether there was an attempt made to share the revised video back with those who offered comments. As such the paper describes the normal process of developing such material. No information is available on whether school administrators were also polled for their views on the videos. The authors note that the effectiveness of the use of the video in improving dental health and literacy is being assessed. Since the paper is merely describing the standard process that is used for development of such material, the authors fail to convey the value of such a paper. Perhaps it may be a better value to include the assessments that they note as being in progress to be added into the paper to make it more complete.

6. PLOS authors have the option to publish the peer review history of their article (what does this mean? ). If published, this will include your full peer review and any attached files.

**Do you want your identity to be public for this peer review?** For information about this choice, including consent withdrawal, please see our Privacy Policy .

Reviewer #1: No

Reviewer #2: **Yes: ** Lakshmi Balaji

---

## [Decision Letter · Decision Letter 1]

2 Mar 2025

PGPH-D-24-02493R1

Development and validation of School Oral Health Promotion Program “SOHEPP” oral health video for the training of adolescents and teachers in Ibadan, Nigeria

Dear Dr. Lawal,

Thank you for submitting your manuscript to PLOS Global Public Health. After careful consideration, we feel that it has merit. We invite you to submit a revised version of the manuscript that addresses the points raised during the review process by reviewer 1.

We look forward to receiving your revised manuscript.

Kind regards,

Maha El Tantawi

Academic Editor

Journal Requirements:

Editor Comments (if provided):

Please address reviewer 1 comments in the attached file. There are about 24 minor comments to address.

Reviewers' comments:

Reviewer's Responses to Questions

**Comments to the Author**

1. If the authors have adequately addressed your comments raised in a previous round of review and you feel that this manuscript is now acceptable for publication, you may indicate that here to bypass the “Comments to the Author” section, enter your conflict of interest statement in the “Confidential to Editor” section, and submit your "Accept" recommendation.

Reviewer #1: (No Response)

Reviewer #2: All comments have been addressed

2. Does this manuscript meet PLOS Global Public Health’s publication criteria ? Is the manuscript technically sound, and do the data support the conclusions? The manuscript must describe methodologically and ethically rigorous research with conclusions that are appropriately drawn based on the data presented.

Reviewer #1: Yes

Reviewer #2: Yes

3. Has the statistical analysis been performed appropriately and rigorously?

Reviewer #1: N/A

Reviewer #2: I don't know

4. Have the authors made all data underlying the findings in their manuscript fully available (please refer to the Data Availability Statement at the start of the manuscript PDF file)?

Reviewer #1: No

Reviewer #2: No

5. Is the manuscript presented in an intelligible fashion and written in standard English?

Reviewer #1: Yes

Reviewer #2: Yes

6. Review Comments to the Author

Reviewer #1: The paper requires minor modifications before it can be accepted for publication. I encourage the authors to use English correction software, such as QuillBot, to enhance their writing in future submissions. Additionally, while some sentences in this paper are long and acceptable, future papers should split them into two for better clarity.

Reviewer #2: The revised manuscript is clearer and describes the validation as the main final step and therefore now addresses all concerns and comments shared earlier.

7. PLOS authors have the option to publish the peer review history of their article (what does this mean? ). If published, this will include your full peer review and any attached files.

**Do you want your identity to be public for this peer review?** For information about this choice, including consent withdrawal, please see our Privacy Policy .

Reviewer #1: No

Reviewer #2: **Yes: ** Lakshmi N Balaji

---

## [Editor Report · Decision Letter 2]

28 Mar 2025

Development and validation of School Oral Health Promotion Program “SOHEPP” oral health video for the training of adolescents and teachers in Ibadan, Nigeria

PGPH-D-24-02493R2

Dear Dr Lawal,

We are pleased to inform you that your manuscript 'Development and validation of School Oral Health Promotion Program “SOHEPP” oral health video for the training of adolescents and teachers in Ibadan, Nigeria' has been provisionally accepted for publication in PLOS Global Public Health.

Best regards,

Maha El Tantawi

Academic Editor
